# Mitigating Quantization Errors Due to Activation Spikes in GLU-Based LLMs

## Abstract

Modern large language models (LLMs) have established state-of-the-art performance through architectural improvements, but still require significant computational cost for inference. In an effort to reduce the inference cost, post-training quantization (PTQ) has become a popular approach, quantizing weights and activations to lower precision, such as INT8. In this paper, we reveal the challenges of activation quantization in GLU variants [40], which are widely used in feedforward network (FFN) of modern LLMs, such as LLaMA family. The problem is that severe local quantization errors, caused by excessive magnitudes of activation in GLU variants, significantly degrade the performance of the quantized LLM. We denote these activations as *activation spikes*. Our further observations provide a systematic pattern of activation spikes: 1) The activation spikes occur in the FFN of specific layers, particularly in the early and late layers, 2) The activation spikes are dedicated to a couple of tokens, rather than being shared across a sequence. Based on our observations, we propose two empirical methods, Quantization-free Module (QFeM) and Quantization-free Prefix (QFeP), to isolate the activation spikes during quantization. Our extensive experiments validate the effectiveness of the proposed methods for the activation quantization, especially with coarse-grained scheme, of latest LLMs with GLU variants, including LLaMA-2/3, Mistral, Mixtral, SOLAR, and Gemma. In particular, our methods enhance the current alleviation techniques (e.g., SmoothQuant) that fail to control the activation spikes.[1]

## 1 Introduction

Large language models (LLMs) have become a key paradigm in natural language processing, accelerating the release of variations within the community [49, 58]. Furthermore, latest LLMs establish state-of-the-art performance by training with increased scale, as well as by adopting architectural improvements such as GLU [40], RoPE [41], GQA [2], and MoE [21]. Especially, GLU (Gated Linear Unit) variants (e.g., SwiGLU, GeGLU) has been adopted in the most of modern LLM architectures (e.g., LLaMA family [46]), due to training efficiency [31, 40]. Although LLMs broaden foundational capabilities in natural language tasks and potential for various applications, billions of parameters in the large models impose considerable computational costs on end users in practice. To reduce GPU memory requirements and accelerate inference speed, post-training quantization (PTQ) offers an affordable solution by quantizing weights and activations into a lower precision (e.g., INT8) without a need for expensive retraining steps [17, 19, 30]. However, recent studies have revealed that large magnitude values at certain coordinates exist in the activations of LLMs, which are often called outliers, posing a key challenge in activation quantization [1, 12, 50, 51]. Another line of works attempts to explain the role of outlier values in the attention mechanism [9, 42]. Nevertheless, current research on the impact of evolving LLM architectures on the outliers remains insufficient.

---

[1]Code is available at `https://anonymous.4open.science/r/activation-spikes-EDF0`.

Submitted to 38th Conference on Neural Information Processing Systems (NeurIPS 2024). Do not distribute.

In this paper, we present our discovery that the GLU architecture in the feed-forward network (FFN) generates excessively large activation values, which are responsible for significant local quantization errors. Specifically, we observe that these problematic activation values occur in specific linear layers and are dedicated to a couple of tokens, which will be discussed in Section 3. To distinguish the excessive GLU activations from the outliers, we refer to them as *activation spikes*. In light of our observations, we propose two empirical methods to mitigate the impact of activation spikes on quantization: Quantization-free Module (QFeM) and Quantization-free Prefix (QFeP). QFeM aims to partially exclude quantization for linear layers (or modules) where large quantization errors occur, instead of quantizing the entire linear modules in the LLM. By scoring the extent of scale disparity, QFeM selects linear modules to exclude. On the other hand, QFeP identifies the prefix that triggers activation spikes and preserves its context as a key-value (KV) cache, thereby preventing the recurrence of activation spikes in subsequent tokens. It is noteworthy that both QFeM and QFeP rely on calibration results to capture activation spikes in advance, without any modifications to the target LLM. This indicates that our methods can be integrated into any existing quantization methods.

In our comprehensive experiments, we demonstrate that recently released LLMs incorporating GLU variants struggle with activation spikes when applying activation quantization. Consequently, the proposed methods, QFeM and QFeP, substantially enhance the performance of the primitive quantization method, the round-to-nearest (RTN) method. Furthermore, we observe that current outlier alleviation methods [50, 51] are exposed to the activation spikes and benefit from our proposed methods. Compared to the strong baseline of fine-grained activation quantization [55], our methods show competitive performance, achieving reduced latency and memory footprint.

In summary, the contributions of our work are as follows:

• We find that the GLU architecture in modern LLMs systematically generates excessive activation values, which are responsible for significant performance degradation in activation quantization.

• Based on our observations, we propose two empirical methods, QFeM and QFeP, which effectively exclude the activation spikes during quantization, with negligible computational overhead and compatibility with any existing quantization techniques.

• Our extensive experimental results validate the detrimental impact of the activation spikes on activation quantization, while our proposed methods consistently enhance the quantization performance.

## 2 Related Works

**Outlier Values in LLMs.** Previously, outlier values have been observed in the transformer-based language models such as BERT [14] and early GPT [36] models through numerous studies [8, 24, 27, 35, 45]. Since the advent of LLMs [10, 57] rooted in the GPT, recent studies by [1, 12, 51] have tackled the existence of outlier values in LLMs. According to them, these outliers exhibit a large magnitude of values at the shared dimensions of hidden states across tokens. More recently, [9, 42] explain that the outliers attribute to the vertical pattern in the attention mechanism [25, 52], which influences the performance of LLMs. In particular, [42] claims a different type of outlier existing in the hidden states of specific tokens. However, prior studies merely focus on the superficial hidden states between the decoder layers. Our work provides a module-level investigation where quantization is applied practically, focusing on different LLM architectures.

**Post-training Quantization for LLMs.** Post-training quantization (PTQ) refers to the quantization of a neural network model to low precision, such as INT8, without additional parameter updates [17, 19]. Especially for LLMs, this approach cost-effectively achieves inference with low memory usage and faster inference latency by quantizing the weights and activations used in matrix multiplication (e.g., linear layer). However, because of the challenges in activation quantization of LLMs, many recent works are mainly focused on the weight-only quantization [11, 13, 15, 23, 26, 39, 54]. Otherwise, the activation quantization faces inherent outliers, which hinder accurate quantization by reducing representation resolution. To address this challenge, [12] proposes a mixed-precision quantization method where the outlier dimensions are computed in high precision. [50, 51] approach migration of scale from activation to weights to alleviate the scale of outlier activations. Along this line of research, we propose to enhance the activation quantization based on our observations.

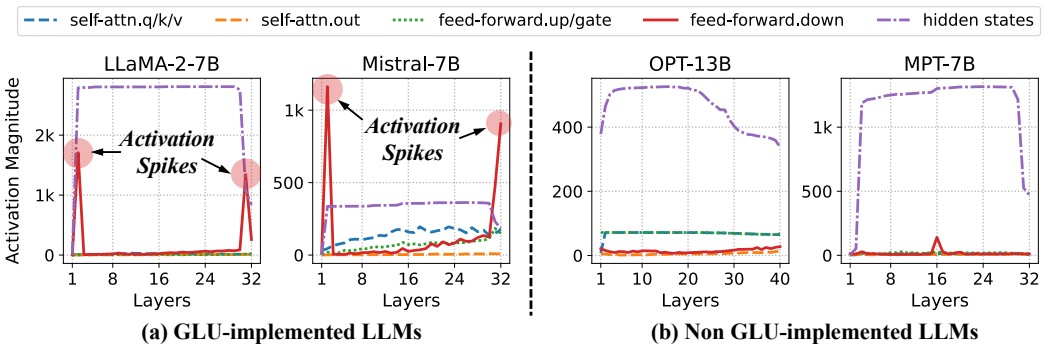

Figure 1: Calibration results on GLU-implemented and non GLU-implemented LLMs. We present the maximum magnitudes of input activations for each linear modules and layer-wise hidden states. For more results on different LLMs, see Appendix A.2, A.3.

## 3 Activation Spikes: Excessive Magnitude of GLU Activations

For clarity, "hidden states" refer to the output tensor of a transformer layer (or block), while "input activations" or "activations" denote the input tensor of a linear layer (or module) in the remain of this paper. Recent work [42] has investigated a novel type of outlier existing in the hidden states across modern LLMs. Although these outliers of hidden states play a crucial role in the attention mechanism [9, 42, 52], their relationship with input activations for quantization has not been fully explored. Importantly, because recent LLMs adopt Pre-LN [4, 53], which normalizes hidden states before self-attention and feed-forward network (FFN) blocks, the scale of hidden states does not reflect the scale of input activations within the transformer block. Therefore, we focus on the input activations fed into each linear module within the transformer block to connect to activation quantization. Specifically, we examine the four linear (projection) layers: `query` (parallel to `key` and `value`), `out`, `up` (parallel to `gate`), and `down` modules. For detailed illustration of Pre-LN transformer, please see Appendix D.1.

### 3.1 Existence of Activation Spikes in GLU Variants

To analyze the input activations, we employ a calibration method, which is used to estimate the quantization factors such as scale and zero-point. For the calibration data, we use 512 samples randomly collected from the C4 [37] training dataset. Afterwards, we feed each sample into the LLM and monitor each hidden state and input activation through the decoder layers. To estimate the scale factor, we use absolute maximum value. The tested LLMs are listed in Appendix A.1.

**GLU-implemented LLMs exhibit activation spikes at specific layers.** In Figure 1a, we display the calibrated scale factors for the LLMs that implement GLU variants (e.g., SwiGLU, GeGLU). Across models, we observe a shared pattern of scale from the results. Within the early and late layers, the `down` modules in the FFN show noticeable magnitudes of input activations. Note that these input activations are derived from the *Hadamard Product* within GLU. Thus, the GLU variants generate activation spikes at the specific layers. Interestingly, we notice a high correlation between the emergence of activation spikes and intermediate hidden states of large scale. This indicates that the FFN contributes to amplifying the hidden states via the addition operation in the residual connection [18]. Once the magnitude of the hidden states is exploded, it persists through layers until encounter the activation spikes at late layers.

**Non GLU-implemented LLMs show modest scale distribution.** Figure 1b illustrates the calibration results for LLMs with the original feed-forward implementation in Transformer [48]. We observe that the LLMs continue to generate the large-scale hidden states, regardless of the GLU implementation. This corresponds to the observations in [42]. More importantly, our module-level results elaborate that the scale of hidden states is not transferable to the input activations of inner linear modules. Instead, we reveal that GLU variants are associated with the hidden states and generate activation spikes. This clarifies the quantization challenge of the GLU-implemented LLMs concentrated in the early and late layers. Because excessive scales of activation spikes have the potential to hinder the accurate quantization, we conduct an in-depth analysis to better understand these activation spikes in the following sections.

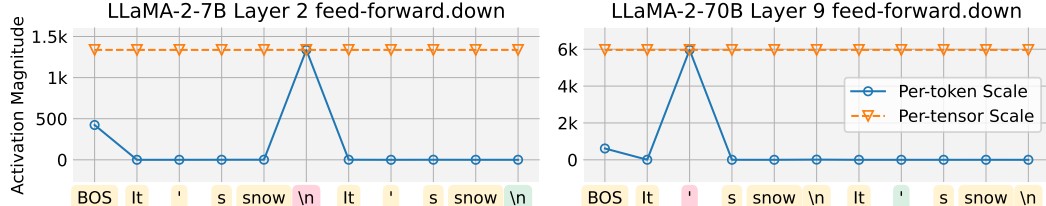

Figure 2: Token-wise scales in a specific layer with an activation spike. When quantizing the input activations using a per-tensor scale, the scale of the activation spike dominates the scales of the other tokens. For more examples, see Appendix D.2.

## 3.2 Token-level Scale Analysis within Activation Spikes

In the previous section, we observed the excessive scale of the input activations derived from GLU activation. When quantizing the input activations, the variance of input activation scales for each token affects the quantization performance [55]. To delve into the disparity between token-wise scales in the activation spikes, we unroll them through the sequence of tokens. Figure 2 illustrates the individual input activation scales where the activation spike appears. Given a token sequence, the large magnitudes of input activations are observed in a couple of tokens, such as the BOS token, newline (\n), and apostrophe ('). These specific tokens coincide with the observations of [42], which suggests that such tokens exhibit massive values in the hidden states. Thus, the activation spike is associated with the process of assigning a special role to these tokens in later transformer layers. However, the excessive scale of specific token hinders the estimation of scale factor for the other tokens, such as in per-tensor quantization. Additionally, the largest scale is dedicated to the first instance of the specified token, while the following usage exhibits a modest scale. This phenomenon makes the quantization more complicated, as the activation spikes dynamically occur depending on the current input sequence.

## 3.3 Effect of Quantization on Activation Spikes

We explore the impact of local quantization errors caused by activation spikes on LLM outputs. To identify the layers where activation spikes occur, we utilize a ratio between the maximum and median values of the token-wise input activation scales, instead of using the maximum scale value alone. The max-median ratio for linear layer $m$ can be formulated as $r^{(m)} = \frac{\max(\mathbf{S}^{(m)})}{\text{median}(\mathbf{S}^{(m)})}$, where $S^{(m)}$ represents the token-wise input activation scales incoming to module $m \in M$. This max-median ratio captures the extent to which maximum scale dominate the other token scales. For comparison, we choose the activation quantization targets as the top-4, middle-4, and bottom-4 modules, based on the max-median ratio in descending order. Then, we evaluate the perplexity and mean-squared error (MSE) using the calibration dataset. Here, the MSE is calculated for the last hidden states between the original (FP16) and partially quantized LLM. As shown in Table 1, quantization on the top-4 rated modules solely degrades the LLM performance by significant margins, while the other cases exhibit negligible performance changes. We consider these quantization-sensitive input activations (*inter alia* activation spikes) to be the quantization bottleneck, which, in this paper, refers to the quantization error caused by outliers.

Furthermore, the activation spikes are conditioned on the specific context of the input sequence as discussed in Section 3.2. Altogether, such dynamic bottlenecks must be handled with caution to enhance the quantization performance of LLMs.

Table 1: Perplexity and MSE of partial activation quantization of LLMs

| Model | Perplexity(↓) | | | | MSE(↓) | | |
|---|---|---|---|---|---|---|---|
| | FP16 | Top 4 | Middle 4 | Bottom 4 | Top 4 | Middle 4 | Bottom 4 |
| LLaMA-2-7B | 7.37 | **11.77** | 7.38 | 7.40 | **1908.80** | 1.03 | 12.90 |
| LLaMA-2-13B | 6.84 | **15.09** | 6.84 | 6.84 | **4762.11** | 0.91 | 10.38 |
| Mistral-7B | 8.35 | **69.45** | 8.35 | 8.36 | **218.60** | 0.02 | 0.18 |
| Gemma-7B | 10.85 | **85.83** | 10.94 | 10.87 | **213.93** | 1.60 | 1.07 |

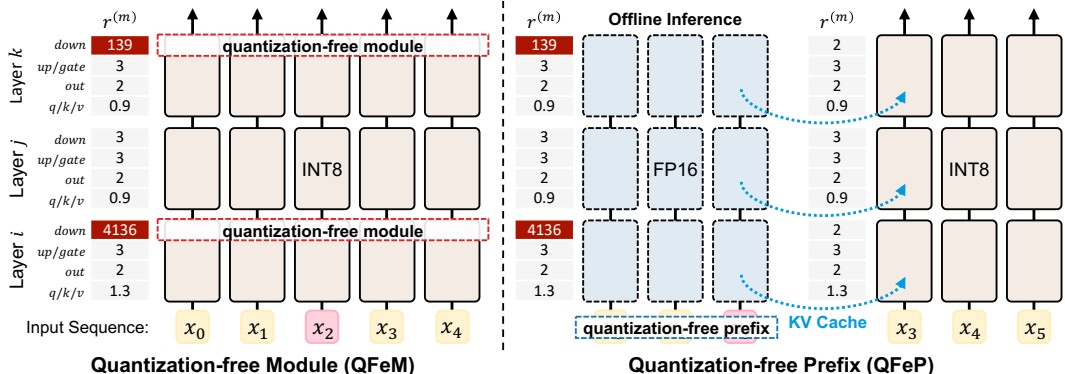

Figure 3: Overview of QFeM and QFeP. (Left): QFeM excludes the modules whose $r^{(m)}$ is larger than the hyperparameter $\alpha$ from quantization. (Right): QFeP computes in advance the prefix of activation spikes and utilizes solely their KV cache during the quantization phase, effectively preventing further activation spikes in subsequent sequences.

## 4 Mitigating Quantization Quality Degradation Based on the Observation

To address the quantization bottleneck, our approach is based on the deterministic occurrence patterns of activation spikes. First, we utilize the observation that bottlenecks occur at a few specific layers. This implies that naive full quantization of LLMs is affected by these bottlenecks. Second, we exploit the phenomenon that the activation spike is derived from the first occurrence of specific tokens. Thus, the planned occurrence prevents recurrence in the subsequent and possibly future tokens. In the following sections, we propose two methods inspired the above insights.

### 4.1 Quantization-free Module (QFeM)

In the full quantization of LLM, all linear layers within the LLM are quantized. Among these linear layers, we propose omitting the quantization of input activations for linear layers where significant quantization errors are caused by activation spikes. To be noted, increasing the number of unquantized modules exhibits a trade-off between the inference latency and the model performance. Thus, determining which module should be quantized (or left unquantized) is crucial to retain the efficacy of quantization. Here, we use the max-median ratio $r^{(m)}$ and define a set of unquantized modules, denoted as $M_{\text{unq}}$, where the ratio $r^{(m)}$ of each linear layer is larger than threshold $\alpha$. For instance, all linear layers in $M$ are quantized if $\alpha = \infty$. For clarity, we treat sibling linear layers, such as query-key-value, as a single linear layer. To control the impact of activation quantization only, we leave the weight parameters in unquantized linear layers as INT8 and dequantize them into FP16 during matrix multiplication with the incoming activations, operating as weight-only quantization.

**Optimizing the threshold $\alpha$.** To calculate the activation scale ratio for each linear layer, we first gather token-wise input activation scales from the calibration examples discussed in Section 3.1. Exceptionally, for FFN experts in the mixture of experts (MoE) architectures like the Mixtral model [21], calibration is performed separately. After determining these ratios, we use binary search to set the threshold value $\alpha$, balancing inference latency and performance degradation. As a metric, we assess performance through perplexity measured on the same calibration examples. For example, the relationship between threshold value $\alpha$ and its impact on performance is depicted in Figure 4, demonstrating how full quantization can degrade performance. Rather than fully quantizing, we identify an optimal threshold by finding the intersection of two performance curves; in Figure 4, this threshold is approximately 16. Details on the QFeM implementation are provided in Table 2.

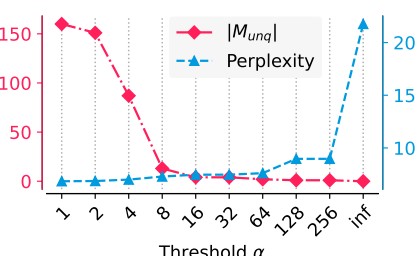

Figure 4: Trade-off between perplexity (stands for performance) and $|M_{unq}|$ (stands for latency) according to the threshold $\alpha$ for LLaMA-2-13B model.

## 4.2 Quantization-free Prefix (QFeP)

Orthogonal to the QFeM, we propose Quantization-free Prefix (QFeP) that mitigates the quantization errors by precomputing the prefix (or short prompt) corresponding to activation spikes. This method is based on the observations presented in Section 3.2, which indicate that significant quantization errors result from the overestimated scale factor of the *first instance* within the restricted token set. Inspired by this occurrence pattern of activation spikes, we aim to construct a prefix which stabilizes the quantization scale factor of the tokens that come after the prefix. In other words, once the prefix is fixed at the beginning, the activation spikes consistently occur within the prefix. Afterward, we employ key-value (KV) caching mechanism to process the activation spikes in advance. In practice, KV cache is utilized to optimize the decoding speed of causal language models by storing precomputed key and value states of the previous tokens [32, 34]. This approach provides a bypass of the quantization including activation spikes, while preserving the context of prefix through the KV cache. The KV cache for the prefix is precomputed once through the offline inference of LLM without quantization. Then, this KV cache is exploited in the quantization phases, such as calibration or dynamic quantization, even for quantized inference. The process of QFeP is illustrated in Figure 3.

**Prefix Search.** To form a prefix of explicit activation spike, we first identify candidate token that represent the activation spike at the linear layer with the highest max-median ratio $r^{(m)}$. For instance, the candidate token can be apostrophe (') token for LLaMA-2-70B model, as highlighted in red in Figure 2. Once the candidate token is identified, we search the middle context token for between the BOS token and the candidate token in the prefix. This middle context provides dummy context, which is required to activate the candidate token. To find the middle context, we design a template $[B, T_1, C_1, T_2, C_2]$ where $B$, $T_i$, and $C_i$ denote the BOS token, context token, and candidate token in the vocabulary $V$, respectively. Then, we select the context token $T$ where $C_1$ triggers an activation spikes, while later instance of the same token $C_2$ does not. When the context token for the activation spikes is varied, we choose the token that maximizes the activation scale ratio between the $C_1$ and $C_2$. Finally, we prepare the KV cache for searched prefix of $[B, T, C]$. Note that the latter sequence in the template can be replaced with sequences from dataset instead of repetition.

**Implementation Details.** During the prefix search phase, we exploit the calibration dataset used in Section 3.1. For the candidate tokens, we consider the tokens with the top three largest input activation magnitudes. Then, we search for the middle context token among top 200 most frequent tokens in the calibration dataset, which is the subset of the vocabulary $V$. Finally, with the search result, we prepare the KV cache for the target model in FP16 precision. Exceptionally, for the Mixtral [21] model, we use the scale of output hidden states instead of input activations, as the tokens are divided sparsely in a mixture of experts architecture. Table 2 presents the searched prefix.

Table 2: Specifications for QFeM and QFeP used in experiments. $|M|$ denotes the total number of linear layers in the LLM, and $|M_{unq}|$ represents the number of unquantized layers for QFeM.

| Model | Prefix | $\alpha$ | $|M_{unq}|/|M|$ |
|---|---|---|---|
| LLaMA-2-7B | [BOS] all . | 6.68 | 17 / 128 |
| LLaMA-2-13B | [BOS] then , | 12.91 | 6 / 160 |
| LLaMA-2-70B | [BOS] I ' | 9.16 | 25 / 320 |
| Mistral-7B | [BOS] how \n | 49.00 | 3 / 128 |
| Mixtral-8x7B | [BOS] ). \n | 4.03 | 191 / 608 |
| SOLAR-10.7B | [BOS] a 1 | 6.48 | 11 / 192 |
| Gemma-7B | [BOS] . Più | 10.65 | 5 / 112 |
| LLaMA-3-8B | [BOS] - nd | 6.64 | 6 / 128 |
| LLaMA-3-70B | [BOS] and , | 78.37 | 3 / 320 |

## 5 Experiments

### 5.1 Experimental Setup

**Models.** Our proposed methods, QFeM and QFeP, aim to mitigate the quantization bottleneck, which is discussed in Section 3.3, caused by the activation spikes, especially in the GLU variants. To validate the efficiency proposed methods, we tested publicly released LLMs that were implemented with GLU, according to their paper and source code. We recognize recent LLMs, including LLAMA-2-{7B, 13B, 70B} [47], LLaMA-3-{7B, 70B}, Mistral-7B [20], Mixtral-8x7B [21], SOLAR-10.7B [22], and Gemma-7B [43], utilize the GLU architecture. The LLMs with original FFN are not covered, as they suffer from the existing outliers rather than activation spikes. All models are sourced from the huggingface-hub[2] repository.

---

[2]https://huggingface.co/models

Table 3: Perplexity and zero-shot evaluation for the quantization on LLaMA-2 models. FP16 denotes the original model precision, and W8A8 denotes the model quantized to INT8 for both weights and activations.

| Method | WikiText-2 (ppl↓) | PIQA (acc↑) | LAMBADA (acc↑) | HellaSwag (acc↑) | WinoGrande (acc↑) | Avg (acc↑) |
|---|---|---|---|---|---|---|
| | | | LLaMA-2-7B | | | |
| FP16 | 5.268 | 78.18% | 73.67% | 57.13% | 69.46% | 69.61% |
| W8A8 | 8.634 | 72.80% | 62.27% | 49.57% | 63.69% | 62.08% |
| +QFeM | 5.758[**-2.876**] | 78.02% | 73.86% | 56.32% | 68.35% | 69.14%[**+7.06**] |
| +QFeP | 5.758[**-2.876**] | 76.44% | 73.57% | 55.55% | 69.22% | 68.69%[**+6.61**] |
| +QFeM+QFeP | 5.573[**-3.061**] | 77.86% | 74.58% | 56.05% | 69.38% | 69.47%[**+7.39**] |
| | | | LLaMA-2-13B | | | |
| FP16 | 4.789 | 79.49% | 76.54% | 60.20% | 72.38% | 72.15% |
| W8A8 | 34.089 | 70.13% | 49.66% | 42.65% | 58.72% | 55.29% |
| +QFeM | 5.241[**-28.848**] | 77.58% | 75.68% | 59.13% | 72.61% | 71.25%[**+15.96**] |
| +QFeP | 6.000[**-28.089**] | 77.53% | 73.94% | 57.23% | 70.96% | 69.91%[**+14.62**] |
| +QFeM+QFeP | 5.126[**-28.963**] | 78.51% | 75.86% | 59.44% | 72.61% | 71.61%[**+16.32**] |
| | | | LLaMA-2-70B | | | |
| FP16 | 3.218 | 81.45% | 79.45% | 65.29% | 80.43% | 76.65% |
| W8A8 | 8.055 | 74.05% | 70.27% | 55.21% | 67.96% | 66.87% |
| +QFeM | 3.830[**-4.225**] | 81.23% | 77.66% | 64.15% | 78.14% | 75.30%[**+8.43**] |
| +QFeP | 6.007[**-2.048**] | 77.64% | 73.26% | 63.40% | 76.16% | 72.62%[**+5.75**] |
| +QFeM+QFeP | 3.708[**-4.347**] | 81.23% | 77.82% | 64.65% | 77.11% | 75.20%[**+8.33**] |

Figure 5: The average accuracy of zero-shot evaluation on other GLU-implemented LLMs. Most models recover significantly compared to W8A8, with performance close to FP16.

**Quantization.** In the experiments, we quantize both the input activations and the weights of linear layers for INT8 matrix multiplication operations. Note that in Table 2, $|M|$ denotes the total number of linear modules targeted for quantization. In these linear layers, we opt for dynamic per-tensor quantization as the quantization scheme of input activations, and per-channel quantization for weights, respectively. Regarding both input activations and weights, we symmetrically quantize the range using the absolute maximum value as the scale estimation function. For comparison, we use FP16 and per-token activation quantization [55] as baselines. We refer the reader to Appendix B for Batch Matrix-Multiplication (BMM) quantization, which involves quantizing tensors in the self-attention.

**Evaluations.** We evaluate the quantized LLMs with two metrics: zero-shot evaluation accuracy and perplexity. For zero-shot evaluation, we use the four datasets: PIQA [7], LAMBADA [33], HellaSwag [56], and WinoGrande [38]. We utilize the lm-evaluation-harness library [16] to evaluate zero-shot tasks. To measure perplexity, we use the WikiText-2 [28] dataset. In all cases, we use the [BOS] token as the starting token for each input sequence by default.

## 5.2 Main Results

**LLaMA-2 Models.** We report the evaluation results of quantization on LLaMA-2 models in Table 3. Compared to FP16 precision, quantizing both weights and activations (W8A8) degrades the overall performance. The results demonstrate that our proposed methods resolve the activation spikes and, surprisingly, restore the performance of the W8A8 close to that of FP16. For example, the LLaMA-2 7B model achieves less than a 1% performance drop from FP16. It is worth noting that the

Table 4: Evaluation of outlier alleviation methods with QFeM and QFeP. We report perplexity on WikiText-2 and averaged accuracy of four zero-shot tasks. The same quantization scheme for used on both SQ and OSP. Per-tensor weight quantization results are provided in Appendix C.1.

| Method | LLaMA-2-7B | | LLaMA-2-13B | | LLaMA-2-70B | |
|---|---|---|---|---|---|---|
| | ppl($\downarrow$) | acc($\uparrow$) | ppl($\downarrow$) | acc($\uparrow$) | ppl($\downarrow$) | acc($\uparrow$) |
| SQ [51] | 9.907 | 61.08% | 34.869 | 59.45% | 8.800 | 70.25% |
| +QFeM | **5.534** | **69.65%** | **5.118** | **71.23%** | **3.599** | **75.93%** |
| +QFeP | 5.715 | 68.66% | 6.551 | 69.33% | 5.228 | 74.07% |
| OSP [50] | 38.490 | 59.90% | 5.148 | 71.29% | 3.827 | 75.52% |
| +QFeM | **5.493** | **69.37%** | **5.099** | **71.37%** | **3.559** | **75.92%** |
| +QFeP | 5.642 | 68.95% | 5.144 | 71.05% | 3.752 | 75.36% |

proposed QFeM and QFeP improve at comparable levels. This indicates that the activation spikes present a direct cause of the significant decrease in quantization performance. Because the proposed methods are orthogonal, the performance slightly increases when incorporating both QFeM and QFeP compared to applying them individually.

**Other GLU-implemented LLMs.** For other LLMs that incorporate GLU, we investigated the effectiveness of our methods in mitigating the quantization bottleneck. As can be seen in Figure 5, our methods consistently remedy the performance drop caused by activation spikes. Noticeably, the Mixtral model demonstrates robustness towards the performance degradation. This indicates that the mixture of experts architecture, which divides the MLP experts by tokens, helps to alleviate the impact of the activation spikes. Meanwhile, addressing the activation spikes is not a sufficient complement for the Gemma model compared to other models. We attribute this to the choice of activation function among GLU variants; specifically, Gemma uses GeGLU, while other models employ SwiGLU.

### 5.3 Combining Outlier Alleviation Methods

While our method focuses on the activation spikes, the inherent outlier values in the input activations remain. Here, we combine the prior outlier alleviation methods, such as SmoothQuant (SQ) [51] and OutlierSuppressionPlus (OSP) [50], to further improve the quantization error. In practice, our methods are utilized during the scale calibration phase of alleviation methods to mitigate the impact of activation spikes on scale migration between activations and weights. Table 4 demonstrates the evaluation results of applying the outlier alleviation methods solely and combining them with our methods. We find that there are cases where the alleviation method fails to recover the performance when quantizing the activations with per-tensor scheme.[3] This indicates that alleviating the outlier scales, including the activation spikes, is challenging. With the QFeM, the activation spikes are excluded, and the accurate alleviation is enabled. In addition, the QFeP also benefits from the SQ method, as seen in the case of LLaMA-2 70B. Exceptionally, the OSP successfully addresses the activation spikes in the 13B and 70B cases.

### 5.4 Ablation Study

For the QFeP, we designed a length-three prefix for the KV cache, including the BOS token, context token, and extra token for activation spike. Because the KV cache consumes the capacity of the pretrained sequence position, it raises a question about the length of the prefix. Therefore, we conduct ablation study for different prefixes for the KV cache. For the pre-

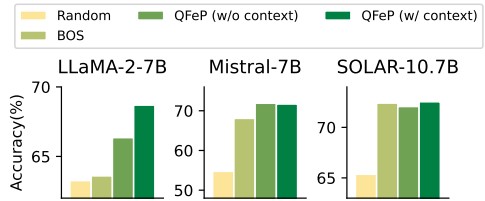

Figure 6: Prefix ablation. Y-axis represents averaged accuracy of four zero-shot tasks.

fixes, we prepare random, BOS only, and both QFeP without and with the context token. We illustrate the results of ablation study in Figure 6. In all cases, the random prefix showcases the lowest performance. While the KV cache with the BOS token demonstrates inconsistent performance, our QFeP

---

[3]In their papers, the activations of LLaMA models are quantized using only a per-token scheme.

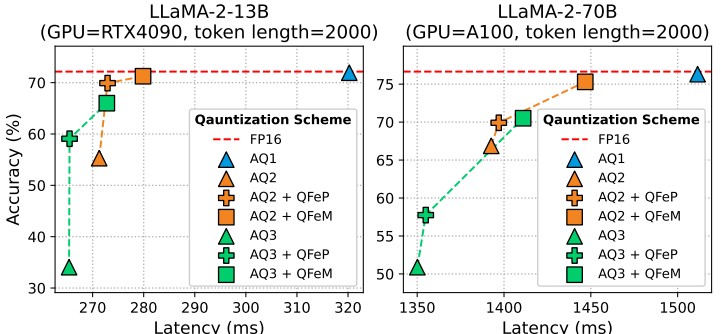

| Method | SeqLen | |
|--------|------|------|
|        | 1K   | 2K   |
| LLaMA-2-7B | | |
| AQ1    | 8185MiB | 9516MiB |
| AQ2    | 8148MiB | 9474MiB |
| +QFeP  | 8149MiB | 9478MiB |
| +QFeM  | 8148MiB | 9474MiB |
| LLaMA-2-70B | | |
| AQ1    | 67756MiB | 69037MiB |
| AQ2    | 67648MiB | 68820MiB |
| +QFeP  | 67651MiB | 68822MiB |
| +QFeM  | 67838MiB | 68819MiB |

Table 5: Memory footprint.

Figure 7: Accuracy-latency comparison of different activation quantization schemes: dynamic per-token (AQ1), dynamic per-tensor (AQ2), and static per-tensor (AQ3).

consistently shows significant improvement. Importantly, the results imply that the sufficient prefix for the models exhibits differences. However, we emphasize that our KV design for QFeP shows improvements by large margins across all models.

## 5.5 Computational Cost Analysis

The proposed methods require additional resources to evict the activation spikes. Therefore, we analyze the computational costs of the methods and compare them in various schemes. For comparison, we evaluate different activation quantization schemes: dynamic per-token, dynamic per-tensor, and static per-tensor, denoted as AQ1, AQ2, and AQ3, respectively. This distinction establishes strong baselines and demonstrates the potential of the methods. To calibrate the static scales, we estimate the absolute maximum value using the calibration dataset, which is used in Section 3.1.

**Inference Latency.** For each setting, we present the accuracy of the zero-shot tasks and inference latency of the fixed token sequence, as shown in Figure 7. While the fine-grained scheme (AQ1) shows a negligible accuracy drop, the counterparts (AQ2, AQ3) degrade with the quantization bottleneck. However, by applying our methods, the coarse-grained schemes achieve a competitive performance gain. For example, the combination of AQ2 and QFeM demonstrates the performance close to the AQ1 but with faster latency. The results signify that addressing the quantization bottleneck is important to accelerate the inference latency with coarser granularity. Specifically, the naive static quantization (AQ3), the fastest scheme, exhibits a significant decline. We hope that our work contributes to the future works, which address the remaining challenges in static quantization.

**Memory Footprint.** In Table 5, we record the maximum memory footprint of our methods. For QFeP, the additional memory is consistently required for the preserved KV cache. However, this memory overhead is much smaller than that used in the fine-grained quantization (AQ1), as QFeM utilizes only three tokens for the cache. Contrary to QFeP, QFeM shows inconsistent memory utilization. For example, the 7B model with QFeM exhibits memory usage similar to AQ2, while the 70B model with QFeM incur additional consumption for a sequence length of 1K. This is attributed to the use of W8A16 for the unquantization modules in QFeM. To tailor the memory usage or inference speed, an alternative strategy can be utilized for QFeM, such as applying fine-grained activation quantization to the unquantization modules instead of using W8A16.

## 6 Conclusion

We explore the quantization challenge of GLU activations for modern LLMs. We find that the GLU variants generates excessive activation scales, which cause significant quantization bottlenecks at the specific layers. Based on the systematic generation pattern of the activation spikes, we propose methods that address the spikes in a layer-wise (QFeM) and token-wise manner (QFeP). In the experiments, we confirm that the proposed methods effectively resolve the quantization bottlenecks and result in a large performance gain. We expect that our work sheds light on the potential challenges in future studies regarding quantization and facilitates the development of efficient LLM systems.

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

## A  Additional Calibration Results

In this section, we provide details of LLMs when performing calibration, which is the step during quantization where the FP16 ranges are computed (Appendix A.1), and additional calibration results (Appendix A.2, A.3).

### A.1  Detailed Specification of LLMs

In Section 3.1, we have performed the calibration method on various LLMs. We observe the calibration results by categorizing based on the presence of GLU in the LLMs. Table 6 shows the detailed structures of the LLMs. We refer notations for feed-forward implementiation from [40]. In the case of GLU-implemented LLMs, which is LLaMA-2, LLaMA-3, Mistral, Mixtral, SOLAR, StableLM-2, and Gemma, most models have SwiGLU for FFN activation, while only Gemma has GeGLU. On the other hand, in non GLU-implemented LLMs, most of them utilize GeLU for FFN activation, with the exception of OPT, which uses ReLU.

Table 6: Architecture specification of LLMs. We categorize them into two groups depending on whether GLU is implemented in the FFN. All LLMs in the table use Pre-LN for the LayerNorm position.

| Model | Size | FFN Activation | Normalization | PE | Vocabulary Size |
|---|---|---|---|---|---|
| *GLU-implemented LLMs:* | | | | | |
| LLaMA-2 [47] | 7B, 13B, 70B | SwiGLU | RMSNorm | RoPE | 32000 |
| LLaMA-3 | 8B, 70B | SwiGLU | RMSNorm | RoPE | 128256 |
| Mistral [20] | 7B | SwiGLU | RMSNorm | RoPE | 32000 |
| Mixtral [21] | 8x7B | SwiGLU | RMSNorm | RoPE | 32000 |
| SOLAR [22] | 10.7B | SwiGLU | RMSNorm | RoPE | 32000 |
| StableLM-2 [5] | 12B | SwiGLU | LayerNorm | RoPE | 100352 |
| Gemma [43] | 7B | GeGLU | RMSNorm | RoPE | 256000 |
| *Non GLU-implemented LLMs:* | | | | | |
| OPT [57] | 6.7B, 13B, 30B, 66B | ReLU | LayerNorm | Learned | 50272 |
| MPT [44] | 7B, 30B | GeLU | LayerNorm | ALiBi | 50432 |
| Pythia [6] | 6.9B, 12B | GeLU | LayerNorm | RoPE | 50432, 50688 |
| Falcon [3] | 7B, 40B | GeLU | LayerNorm | RoPE | 65024 |
| Phi-2 [29] | 2.7B | GeLU | LayerNorm | RoPE | 51200 |

### A.2  Other Calibration Results on GLU-implementation

Figure 8, 9 show the calibration result examples for various GLU-implemented LLMs that are not shown in the models in Figure 1a. In most GLU-implemented LLMs, we observe that the input activations have large values near the first and last layers. Unlike the typical GLU-implemented LLM architecture, Mixtral is composed of 8 feed-forward blocks in the single FFN, containing multiple gate linear units [21]. According to this structure, we can observe that one of the gates spikes in value in Figure 8.

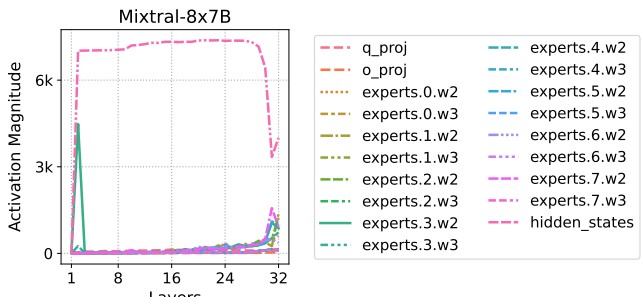

Figure 8: **Calibration results on GLU-implemented LLMs (Mixtral-8x7B).**

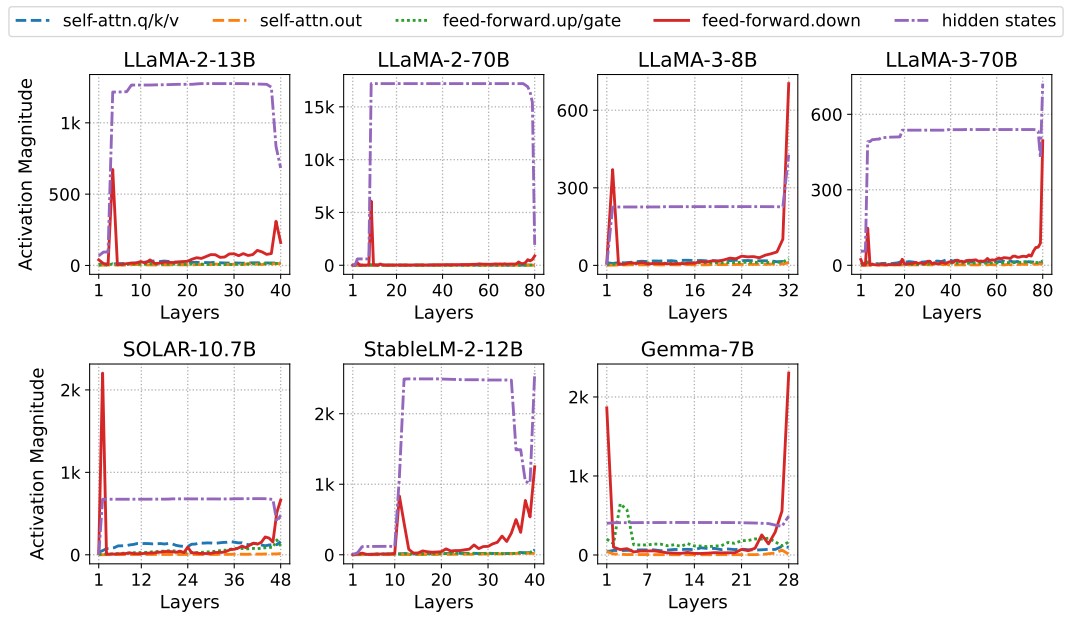

Figure 9: **Calibration results on GLU-implemented LLMs.**

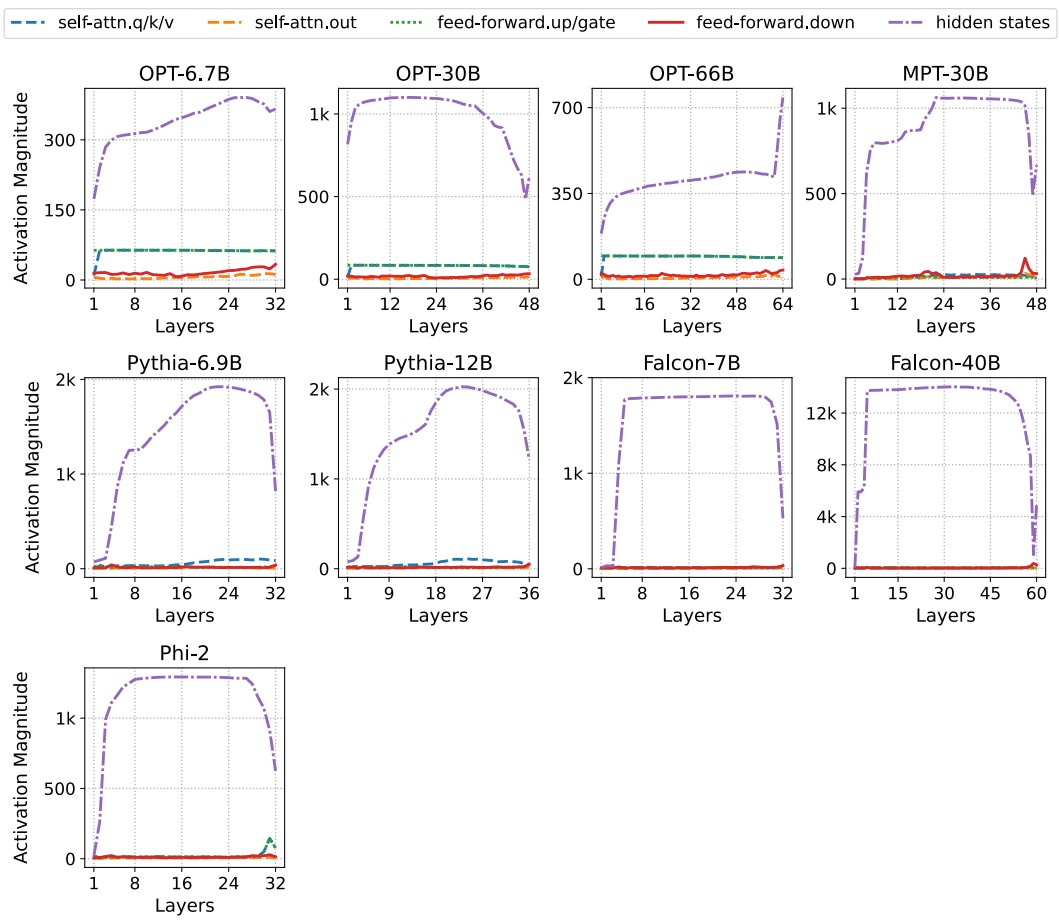

Figure 10: **Calibration results on Non GLU-implemented LLMs.**

### A.3 Other Calibration Results on Non GLU-implementation

Figure 10 shows the calibration result examples for various non GLU-implemented LLMs that were not shown in the models in Figure 1b. There are no activation spikes on non GLU-implemented LLMs.

# B  BMM Quantization

To achieve faster inference latency, BMM operations in the self-attention also can be computed as INT8 operation [51]. This requires a quantization on the query, key, and value states including the cached context. Because activation spikes produce a large magnitude of latent values, it is important to confirm the extent of quantization errors from KV quantization. This confirmation is necessary to gain advantages from BMM quantization. In Table 7, we examine the impact of BMM quantization on the W8A8 and QFeM. Regardless of the BMM quantization, the QFeM method consistently improves the quantization bottleneck. For example, the 13B and 70B models maintain their performance, while the 7B model shows a slight decrease. However, this decrease appears to be due to inherent quantization errors rather than a quantization bottleneck from activation spikes. As a result, we confirm that our QFeM method effectively improves the overall performance even in the BMM quantization scenario.

Table 7: BMM quantization results.

| Model | Method | BMM Quantization | |
|---|---|---|---|
| | | No | Yes |
| 7B | W8A8 | 62.08% | 61.66% |
| | +QFeP | 68.69% | 68.30% |
| 13B | W8A8 | 55.29% | 55.43% |
| | +QFeP | 69.91% | 69.77% |
| 70B | W8A8 | 66.87% | 66.75% |
| | +QFeP | 72.62% | 72.69% |

# C  Supplementary Experiment Results

### C.1 Additional Results for Combining Outlier Alleviation Methods

In Table 8, we provide additional results for Section 5.3 with coarse-grained quantization (i.e., per-tensor quantization) scheme for weight quantization. Compared to the results obtained with per-channel weight quantization in Table 4, these results elucidate the negative impact of activation spikes on the performance of outlier alleviation methods. Furthermore, this suggests that the performance of OSP method resort to the weight quantization scheme. Nevertheless, the proposed methods, QFeM and QFeP, consistently improve the effectiveness of outlier alleviation methods by mitigating the impact of activation spikes.

Table 8: Evaluation of outlier alleviation methods with QFeM and QFeP. We report perplexity on WikiText-2 and averaged accuracy of four zero-shot tasks. Compared to Table 4, per-tensor weight quantization and dynamic per-tensor activation quantization are used.

| Method | LLaMA-2-7B | | LLaMA-2-13B | | LLaMA-2-70B | |
|---|---|---|---|---|---|---|
| | ppl($\downarrow$) | acc($\uparrow$) | ppl($\downarrow$) | acc($\uparrow$) | ppl($\downarrow$) | acc($\uparrow$) |
| SQ [51] | 24.661 | 56.87% | 120.966 | 53.06% | 8.435 | 67.08% |
| +QFeM | **6.016** | **67.74%** | **5.464** | **70.04%** | **4.015** | **74.18%** |
| +QFeP | 6.122 | 67.22% | 10.473 | 68.17% | 5.998 | 72.54% |
| OSP [50] | 9.131 | 63.61% | 8.997 | 64.03% | 6.492 | 71.13% |
| +QFeM | 5.951 | **68.65%** | **5.284** | **70.67%** | **4.434** | 73.30% |
| +QFeP | **5.821** | 68.25% | 5.868 | 67.96% | 4.976 | **73.57%** |

## D Miscellaneous

### D.1 Transformer Architecture.

In Figure 11, we illustrate the Pre-LN transformer architecture and each sub-modules. We highlight with the same color the linear modules that accept identical input activations. Note that the hidden states are normalized before forwarding into the `query` and `up` linear modules.

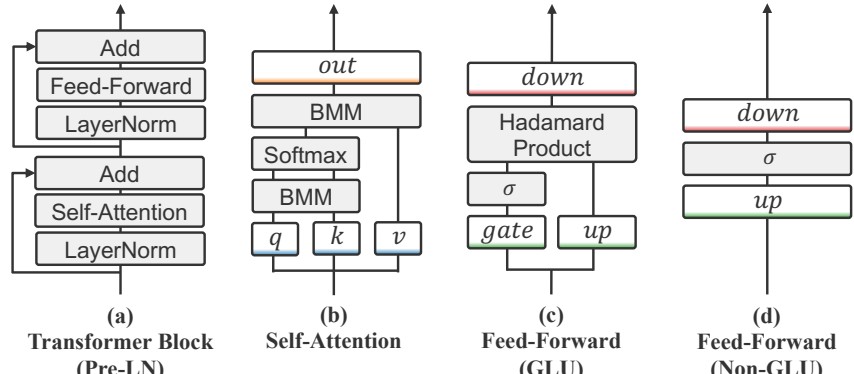

|(a) | (b) | (c) | (d) |
| :---: | :---: | :---: | :---: |
| **Transformer Block (Pre-LN)** | **Self-Attention** | **Feed-Forward (GLU)** | **Feed-Forward (Non-GLU)** |

Figure 11: An illustration of Pre-LN transformer block and its sub-modules. Two feed-forward implementation, GLU and Non-GLU, are visualized in (c) and (d) respectively. In feed-forward network, $\sigma$ denotes non-linear activation function, such as GeLU. We highlight the linear modules where input activations are quantized.

### D.2 Additional Results for Token-level Scale Analysis

We provide additional results for token-level scale analysis (Section 3.2). In Figure 12 and Figure 13, the token for the activation spikes behind the BOS token does not exhibit the excessive activation scale.

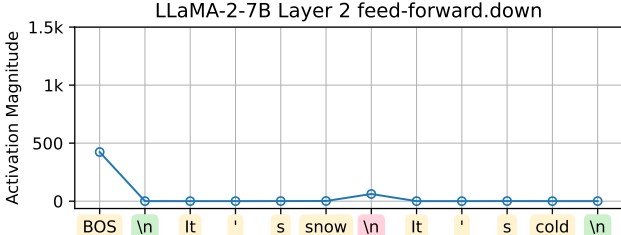

Figure 12: Token-wise scales analysis for LLaMA-2-7B. The newline token behind the BOS token does not exhibit the activation spikes.

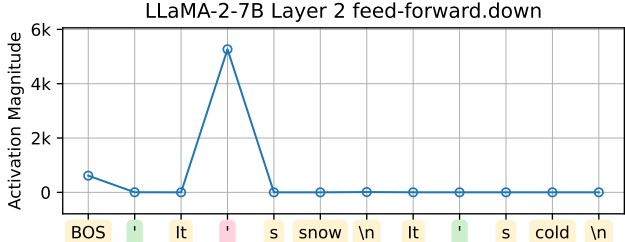

Figure 13: Token-wise scales from the unrolled activation spike of LLaMA-2-70B. The newline token behind the BOS token does not exhibit the activation spikes.

