# OpenReview forum: "Mitigating Quantization Errors Due to Activation Spikes in GLU-Based LLMs"
_NeurIPS.cc/2024/Conference — Submitted to NeurIPS 2024_

### Official Review · Reviewer_PpZ5 · 2024-07-03

**Soundness:** 3
**Presentation:** 2
**Contribution:** 2
**Rating:** 5
**Confidence:** 4

**Summary:**

This paper pays attention to extremely large outliers in LLMs and further investigates the reasons behind these "attention spikes." Consequently, the authors propose two methods to enhance the performance of quantized models.

**Strengths:**

1. The analysis of attention spikes is thorough and comprehensive.

2. The exploration of the relationship between attention spikes and Gated Linear Units (GLU) variants is both interesting and insightful.

**Weaknesses:**

1. The proposed QFeM method is not hardware-friendly, as it maintains some modules at high precision and cannot directly utilize low-bit INT General Matrix Multiply (GEMM) for activations and weights.

2. The proposed QFeP method bears a strong resemblance to a previously researched method, IntactKV[1], yet lacks a detailed comparative discussion.

3. The experimental settings are limited to W8A8 configurations, which previous research, such as SmoothQuant[2], has shown can nearly achieve lossless quantization for W8A8 models.

4. The authors have not included comparisons with state-of-the-art baselines, such as OmniQuant[3], AffineQuant[4], QLLM[5], and QuaRot[6].


[1]. Liu, Ruikang, et al. "IntactKV: Improving Large Language Model Quantization by Keeping Pivot Tokens Intact." arXiv preprint arXiv:2403.01241 (2024).

[2]. Xiao, Guangxuan, et al. "Smoothquant: Accurate and efficient post-training quantization for large language models." International Conference on Machine Learning. PMLR, 2023.

[3]. Shao, Wenqi, et al. "Omniquant: Omnidirectionally calibrated quantization for large language models." arXiv preprint arXiv:2308.13137 (2023).

[4]. Ma, Yuexiao, et al. "Affinequant: Affine transformation quantization for large language models." arXiv preprint arXiv:2403.12544 (2024).

[5]. Liu, Jing, et al. "Qllm: Accurate and efficient low-bitwidth quantization for large language models." arXiv preprint arXiv:2310.08041 (2023).

[6]. Ashkboos, Saleh, et al. "Quarot: Outlier-free 4-bit inference in rotated llms." arXiv preprint arXiv:2404.00456 (2024).

**Questions:**

1. Could you provide a detailed analysis highlighting the differences between the QFeP method and IntactKV?

2. Could you expand the experimental results to include different quantization settings such as W4A4 and W4A8?

3. Could you offer a more detailed comparison with state-of-the-art (SOTA) baselines or conduct the ablation tests as outlined in Table 4?

If the authors can provide more comprehensive results, I am prepared to raise my evaluation scores.

**Limitations:**

Yes.

---

> ### Author Rebuttal · Authors · 2024-08-07
>
> We are deeply grateful for your thorough feedback of our work!
> We hope our response sufficiently addresses your questions.
>
> **Q1.** The proposed QFeM method is not hardware-friendly, as it maintains some modules at high precision and cannot directly utilize low-bit INT General Matrix Multiply (GEMM) for activations and weights.
>
> **A1.** Thank you for your careful comment regarding the hardware side. As you commented, we designed our QFeM to leave an activation tensor in high precision (i.e., FP16), which causes incompatibility with low-bit operations that are supported by hardware level. In such scenario, we encourage applying fine-grained quantization or more advanced quantization (e.g., fine-grained group quantization in Atom [1]) to the target modules ($|M_{unq}|$) of QFeM, as an alternative to leaving them unquantized. This is motivated by QFeM's approach of searching for quantization-sensitive modules using the max-median ratio, as shown in Section 4.1.
>
> [1] Zhao, Yilong, et al. "Atom: Low-bit quantization for efficient and accurate llm serving." Proceedings of Machine Learning and Systems 6 (2024): 196-209.
>
> ---
>
> **Q2.** Could you provide a detailed analysis highlighting the differences between the QFeP method and IntactKV?
>
> **A2.**
> We appreciate your recommendation of the valuable related work of IntactKV [2].
> We carefully read the paper and agree with your comment that our QFeM method strongly resembles IntactKV.
>
> However, there are major differences between QFeP and IntactKV:
> 1. **We identify the activation spikes.**
> Our comprehensive analysis of activation spikes reveals that the large activation scale of activation spikes is responsible for significant degradation of quantization performace. Furthermore, the activation spikes dynamically occur depending on the current input sequence.
> 2. **QFeP addresses all attention spikes.**
> Figure 2 illustrates the activation spikes given a token sequence. Our QFeP searches for dynamic activation spikes using calibration and stores the searched token in the prefix, which prevents recurrence of activation spikes in the subsequent tokens. However, IntactKV includes only the [BOS] token.
> 3. **Prefix ablation study confirmed the efficacy of QFeP.**
> In Section 5.4, we conduct a prefix ablation study for QFeP. Compared to the prefix with only the [BOS] token (which can be viewed as IntactKV), QFeP consistently shows significant improvement through various LLMs.
>
> We acknowledge your recommendation. We intend to include discussions of IntactKV in the final version of our paper.
>
> [2] Liu, Ruikang, et al. "IntactKV: Improving Large Language Model Quantization by Keeping Pivot Tokens Intact." arXiv preprint arXiv:2403.01241 (2024).
>
> ---
>
> **Q3.** The experimental settings are limited to W8A8 configurations, which previous research, such as SmoothQuant, has shown can nearly achieve lossless quantization for W8A8 models.
>
> **A3.** We expand our experiments for low-bit quantization. Please refer to Q2 in the general response. To further our discussion about SmoothQuant, we recommend referring to Q1 in the general response.
>
> ---
>
> **Q4.** Could you offer a more detailed comparison with state-of-the-art (SOTA) baselines or conduct the ablation tests as outlined in Table 4?
>
> **A4.** Thank you for the paper list. Please refer to Q3 in the general response.

---

> ### Comment · Reviewer_PpZ5 · 2024-08-10
> **Minor questions about rebuttal**
>
> I appreciate the authors' discussion and the additional evaluation results provided. However, I have a few minor points that require further clarification. Firstly, IntactKV pre-saves all system prompts as pivot tokens for Vicuna models, which includes all attention spikes, similar to QFeP. Secondly, I am curious as to why QFeP does not improve the performance of the 4-bit LLaMA2-13B model in Appendix 3. Could this be due to the influence of an additional prefix affecting the model's performance?

---

> ### Author Response · Authors · 2024-08-12
>
> We are sincerely grateful for your effort and time in reviewing our rebuttal and for your valuable questions.
>
> ---
>
> **Q5.** IntactKV pre-saves all system prompts as pivot tokens for Vicuna models, which includes all attention spikes, similar to QFeP.
>
> **A5.**
> The discussion continues from our previous response to your Q2, which highlighted the differences between QFeP and IntactKV [1].
>
> Notably, IntactKV selects different pivot tokens depending on the type of LLMs.
> - For pre-trained LLMs (e.g., LLaMA), IntactKV selects only the [BOS] token as a pivot token.
> - For supervised fine-tuned LLMs (e.g., Vicuna), IntactKV stores all system prompts as pivot tokens.
>
> We denote these methods as IntactKV[B] and IntactKV[P], respectively, according to [1].
>
> In our previous rebuttal, we compared QFeP with IntactKV[B] due to the absence of system prompts for pre-trained LLMs.
> When quantizing activations for pre-trained LLMs, QFeP's ability to address activation spikes more effectively than IntactKV becomes evident, as discussed in the previous response.
>
> Nevertheless, further investigation into activation spikes for supervised fine-tuned (or instruction fine-tuned) LLMs would be valuable.
> Previous works have found that pivot tokens persist after instruction fine-tuning [1, 2].
> This finding implies that activation spikes are transferable and that our methods are also effective.
> Indeed, we observed equivalent activation spikes in both LLaMA-2 and its fine-tuned models, such as Vicuna-v1.5 and LLaMA-2-Chat.
> While IntactKV[P] may store long system prompts for pivot tokens, QFeP saves a more compact prefix with a length of 3.
> Furthermore, QFeP is explicit and prompt-agnostic, providing flexibility for system prompts.
> Finally, **QFeP can be applicable to more generalized and effective solutions for addressing activation spikes when applying activation quantization, regardless of whether the LLM is pre-trained or fine-tuned.**
>
>
> [1] Liu, Ruikang, et al. "IntactKV: Improving Large Language Model Quantization by Keeping Pivot Tokens Intact." arXiv preprint arXiv:2403.01241 (2024).
>
> [2] Sun, Mingjie, et al. "Massive activations in large language models." arXiv preprint arXiv:2402.17762 (2024).
>
> ---
>
> **Q6.** I am curious as to why QFeP does not improve the performance of the 4-bit LLaMA2-13B model in Appendix 3. Could this be due to the influence of an additional prefix affecting the model's performance?
>
> **A6.**
> During the preparation of the rebuttal to reviewer geCr, we confirmed that introducing only an additional prefix of QFeP slightly degrades the performance of the FP16 model.
>
> Based on this observation, we hypothesize the following regarding QFeP's contribution to the 4-bit LLaMA-2-13B:
>
> - Case 1 (Atom [3] + QFeP): Given that quantization errors have been minimized via the fine-grained group quantization scheme of Atom, the prepended prefix of QFeP has the possibility to slightly degrade performance. Note that the fine-grained group quantization utilizes a finer granularity than per-token quantization.
> - Case 2 (OmniQuant [4] + QFeP): At present, we are unable to determine the specific factor that degrades the perplexity for WikiText-2, although QFeP achieved a performance gain for C4.
> To provide further evaluation results, we assessed four zero-shot tasks, as shown in the table below.
> The results indicate that QFeP improves the OmniQuant baseline for three tasks, specifically the WinoGrande task.
> This is similar to the results of W4A6 LLaMA-2-13B in Table 2 of the attached PDF.
>
> While there are numerous potential factors that could explain the degradation (e.g., trainable components of OmniQuant or softmax quantization in low-bit), we are making an effort to include an ablation study, similar to Figure 6, to clarify the influence of an additional prefix. Thanks again for your questions.
>
> | Model | #Bits | Method | PIQA($\uparrow$) | LAMBADA($\uparrow$) | HellaSwag($\uparrow$) | WinoGrande($\uparrow$) | Avg($\uparrow$) |
> |---|---|---|:---:|:---:|:---:|:---:|:---:|
> |LLaMA-2-13B|FP16| - | 79.49% | 76.54% | 60.20% | 72.38% | 72.15% |
> |LLaMA-2-13B|W4A4|Atom|78.02%|75.80%|58.48%|70.56%|70.72%|
> |LLaMA-2-13B|W4A4|$\quad$+QFeM|78.07%|75.43%|**58.49%**|70.40%|70.60%|
> |LLaMA-2-13B|W4A4|$\quad$+QFeP|76.99%|**76.19%**|58.22%|70.80%|70.55%|
> |LLaMA-2-13B|W4A4|$\quad$+QFeM+QFeP|**78.24%**|75.61%|58.34%|**71.11%**|**70.83%**|
> |LLaMA-2-13B|W4A4|OmniQuant|67.25%|41.63%|44.95%|52.57%|51.60%|
> |LLaMA-2-13B|W4A4|$\quad$+QFeM|**71.11%**|**48.01%**|46.86%|57.14%|**55.78%**|
> |LLaMA-2-13B|W4A4|$\quad$+QFeP|68.93%|38.99%|45.95%|**58.48%**|53.09%|
> |LLaMA-2-13B|W4A4|$\quad$+QFeM+QFeP|70.02%|44.89%|**46.93%**|57.46%|54.82%|
>
> [3] Zhao, Yilong, et al. "Atom: Low-bit quantization for efficient and accurate llm serving." Proceedings of Machine Learning and Systems 6 (2024): 196-209.
>
> [4] Shao, Wenqi, et al. "Omniquant: Omnidirectionally calibrated quantization for large language models." arXiv preprint arXiv:2308.13137 (2023).

---

> > ### Comment · Reviewer_PpZ5 · 2024-08-12
> >
> > Thanks for your detailed response and additional experiments. Some of my concerns are addressed and I have decided to raise my score.

---

> > > ### Author Response · Authors · 2024-08-12
> > >
> > > Thank you for your insightful reviews of our work and rebuttal during the review period. Your valuable feedback is greatly appreciated and will be incorporated into the final version.

---

### Official Review · Reviewer_TX3p · 2024-07-08

**Soundness:** 2
**Presentation:** 3
**Contribution:** 3
**Rating:** 5
**Confidence:** 3

**Summary:**

This paper identifies some of the underlying causes for why activation quantization (PTQ) could lead to low performance and suggests some methods to address these issues.

**Strengths:**

Please see the “Questions” section.

**Weaknesses:**

Please see the “Questions” section.

**Questions:**

My review is as follows:

1) The results of Table 2 seem to suggest that SmoothQuant leads to an unacceptably high performance degradation. Table 6 of the SmoothQuant paper however shows that for Llamav1, the degradation is very small. I wonder if a difference in implementation is causing this discrepancy in results. I’m sorry if this is mentioned somewhere in the text and I missed it. Could you please clarify?

2) What is the latency for FP16 in Figure 7?

3) There are multiple recent works that suggest quantizing the weights to 4 bits can be done without a big impact to the accuracy. Since memory access (for weights) is typically the bottleneck when LLMs are deployed, it is possible that one may prefer W4A16 over W8A8. In my opinion, studying 4-bit weight quantization (in addition to the already studied W8 quantization) could make the paper more interesting.

4) QFeM method is essentially a mixed precision approach. A lot of quantization papers actually employ some form of mixed precision (even though it is sometimes only mentioned in the footnote). For instance, SmoothQuant uses FP16 for LayerNorm. Has this been taken into account when comparing against other methods? Also, have similar assumptions been made in the implementation of the proposed method?

Minor:

5) There is a type on line 90: “in the remain(der) of this 91 paper”
6) Typo in line 147 “dominate(s)”

Readability and presentation: The paper is mostly easy to understand. One thing I could say is that the idea of QFeM is easier to follow than QFeP; the introduction of the QFeP method may need to be improved and made more concise.

**Limitations:**

Yes.

---

> ### Author Rebuttal · Authors · 2024-08-07
>
> We sincerely thank you for taking the time to review our work!
> We reviewed our paper and fixed some typos, including your suggestions to improve the readability and presentation.
>
> ---
>
> **Q1.** The results of Table 2 seem to suggest that SmoothQuant leads to an unacceptably high performance degradation. Table 6 of the SmoothQuant paper however shows that for Llamav1, the degradation is very small. I wonder if a difference in implementation is causing this discrepancy in results. I’m sorry if this is mentioned somewhere in the text and I missed it. Could you please clarify?
>
> **A1.**
> Thank you for your careful consideration.
> Please refer to Q1 in the general response.
>
> ---
>
> **Q2.** What is the latency for FP16 in Figure 7?
>
> **A2.**
> In Figure 7, we omit the latency of FP16 because LLaMA-13B and LLaMA-70B are too large to deploy into their respective target GPUs (RTX 4090 and A100).
> Instead, we provide the latency for multi-GPU setups.
> The latency for LLaMA-2-13B is approximately 553.06 ms using two RTX 4090 GPUs, while for LLaMA-2-70B, it is around 1673.63 ms using two A100 GPUs.
>
> ---
>
> **Q3.** There are multiple recent works that suggest quantizing the weights to 4 bits can be done without a big impact to the accuracy. Since memory access (for weights) is typically the bottleneck when LLMs are deployed, it is possible that one may prefer W4A16 over W8A8. In my opinion, studying 4-bit weight quantization (in addition to the already studied W8 quantization) could make the paper more interesting.
>
> **A3.**
> Please refer to Q2 in the general response. We expanded our experiment on low-bit quantization.
>
> ---
>
> **Q4.** QFeM method is essentially a mixed precision approach. A lot of quantization papers actually employ some form of mixed precision (even though it is sometimes only mentioned in the footnote). For instance, SmoothQuant uses FP16 for LayerNorm. Has this been taken into account when comparing against other methods? Also, have similar assumptions been made in the implementation of the proposed method?
>
> **A4.**
> In our experiments, we only quantize the linear modules to utilize efficient INT8 matrix multiplication operation.
> In such case, the other modules (e.g., LayerNorm, EmbeddingLayer) use FP16 precision.
> We implement baseline methods (SmoothQuant and OSP) following the same setting.

---

> > ### Comment · Reviewer_TX3p · 2024-08-12
> >
> > Thank you for the detailed responses! It would be great the findings reported in the rebuttal could be explicitly incorporated in the revision. I'll increase my rating from 4 to 5 since some of my concerns regarding the numerical results are already addressed.

---

> > > ### Author Response · Authors · 2024-08-12
> > >
> > > We greatly appreciate your constructive response to our rebuttal! We are highly encouraged by your thoughtful feedback. Thank you.

---

### Official Review · Reviewer_geCr · 2024-07-10

**Soundness:** 3
**Presentation:** 3
**Contribution:** 2
**Rating:** 5
**Confidence:** 3

**Summary:**

This paper addresses the precision challenges posed by the large language models (LLMs) quantization during inference, specifically focusing on the quantization errors in GLU-based feedforward networks. The authors identify that GLU variants in LLMs cause significant local quantization errors due to excessive activation magnitudes, referred to as activation spikes. They observe that GLU-implemented models have larger spikes than non-GLU-implemented models.  They propose two methods, Quantization-free Module (QFeM) and Quantization-free Prefix (QFeP), to isolate and mitigate these spikes during quantization. QFeM leave some linear layers unquantized (usually those layers that cause large activation spikes in the first several layers), and QFeP introduce an additional prefix before the inference process. Their extensive experiments show that these methods improve quantization performance and are compatible with existing techniques.

**Strengths:**

1. The identification of activation spikes in GLU-based LLMs is novel.
2. The paper is well-structured and clear.
3. The QFeP method is novel,  and is somehow similar to the finding of "sink token" in StreamLLM [1].

[1] Xiao, Guangxuan, et al. "Efficient streaming language models with attention sinks." arXiv preprint arXiv:2309.17453 (2023).

**Weaknesses:**

1. My major concern is about the baseline of SmoothQuant reported in Table 4. For example, In Table 7 of SmoothQuant's original paper, they report that W8A8 SQ's PPL of Llama-7B on WikiText-2 dataset is 5.515, while the authors report a PPL of 9.907 on the same dataset. Is there a specific reason about this large gap?

2. In Table 3, the improvement brought by the QFeP method does not seem significant, especially when combining with the QFeM method.

**Questions:**

1. In the method of Quantization-free Prefix (QFeP), will the additional prefix decrease or increase the accuracy? That means you use the QFeP method but do not apply quantization, only introduce the additional prefix.

2. Where do the spikes of the GLU-based model come from in the element-wise multiplication? Do they mostly comes from the gate projection part, or the up projection part?

---

> ### Author Rebuttal · Authors · 2024-08-07
>
> We deeply appreciate the invaluable feedback provided by the reviewer.
>
> ---
>
> **Q1.** My major concern is about the baseline of SmoothQuant reported in Table 4. For example, In Table 7 of SmoothQuant's original paper, they report that W8A8 SQ's PPL of Llama-7B on WikiText-2 dataset is 5.515, while the authors report a PPL of 9.907 on the same dataset. Is there a specific reason about this large gap?
>
> **A1.**
> Please refer to Q1 in the general response.
>
> ---
>
> **Q2.** In Table 3, the improvement brought by the QFeP method does not seem significant, especially when combining with the QFeM method.
>
> **A2.**
> Our proposed two methods both improve the quantization performance by significant margins.
> In the case of the LLaMA-2-7B model with the W8A8 quantization setting, applying QFeM achieves an average zero-shot evaluation accuracy of 69.14%, while applying QFeP achieves 68.69%. When we apply the two methods at the same time, it achieves 69.47%.
>
> The combination of QFeM and QFeP achieves a notable performance boost, showing up to a 29.2% increase when quantizing LLaMA-2-13b to W4A6, as shown in Table 2 (row W4A6+QFeM+QFeP) of the attached PDF.
>
> ---
>
> **Q3.** In the method of Quantization-free Prefix (QFeP), will the additional prefix decrease or increase the accuracy? That means you use the QFeP method but do not apply quantization, only introduce the additional prefix.
>
> **A3.** Thank you for your insightful question! We provide the extra experiment results as below:
>
> |    Model    |       Method       | WikiText-2($\downarrow$) | PIQA($\uparrow$) | LAMBADA($\uparrow$) | HellaSwag($\uparrow$) | WinoGrande($\uparrow$) | Avg($\uparrow$) |
> |:-----------:|:------------------:|:------------------------:|:----------------:|:-------------------:|:---------------------:|:----------------------:|:---------------:|
> |  LLaMA-2-7B |        FP16        |           5.268          |      78.18%      |        73.67%       |         57.13%        |         69.46%         |      69.61%     |
> |             | FP16 (+add Prefix) |           5.281          |      77.53%      |        74.42%       |         56.46%        |         69.85%         |      69.57%     |
> | LLaMA-2-13B |        FP16        |           4.789          |      79.49%      |        76.54%       |         60.20%        |         72.38%         |      72.15%     |
> |             | FP16 (+add Prefix) |           4.800          |      78.84%      |        76.48%       |         60.00%        |         72.30%         |      71.91%     |
>
> The results indicate that our QFeP degrades the performance of the FP16 model slightly. However, our proposed methods improve the performance by significant margins when quantizing the LLMs.
>
> ---
>
> **Q4.** Where do the spikes of the GLU-based model come from in the element-wise multiplication? Do they mostly comes from the gate projection part, or the up projection part?
>
> **A4.**
> At the layer where the activation spikes occur (e.g., Layer 2 FFN), the down projection faces large-scale activations derived from element-wise multiplication. In our analysis, we observe that both tensors (the output activation from the gate projection and the output activation from the up projection) incoming to the multiplication operation contain spikes in the same dimensions.

---

> > ### Comment · Reviewer_geCr · 2024-08-08
> >
> > The results are very interesting.  I increase my score accordingly.

---

> > > ### Author Response · Authors · 2024-08-09
> > >
> > > We appreciate your review of our rebuttal. We're pleased that our results addressed your concerns and improved your assessment of our work.

---

### Official Review · Reviewer_BgAS · 2024-07-11

**Soundness:** 2
**Presentation:** 2
**Contribution:** 2
**Rating:** 5
**Confidence:** 2

**Summary:**

This paper introduces activation quantization methods for GLU-based LLMs, which often face challenges due to activation spikes. To effectively manage these spikes and enable activation quantization using a PTQ-based approach, the paper proposes a Quantization-free Module (QFeM) and a Quantization-free Prefix (QFeP). Specifically, QFeM aims to partially bypass quantization for linear layers where large quantization errors occur. QFeP identifies the prefix that triggers activation spikes and preserves its context as a key-value (KV) cache, preventing the recurrence of activation spikes in subsequent tokens. The paper presents extensive experimental results to compare the accuracy of the quantized models.

**Strengths:**

1) This paper is well organized and easy to understand.
2) The proposed QFeM and QFeP effectively mitigate the impact of activation spikes on activation quantization, preserving the accuracy of LLMs even when activation quantization is applied.
3) The ablation study thoroughly examines the effects of QFeM and QFeP, providing valuable insights.

**Weaknesses:**

1) The perplexity/accuracy results of the baseline methods deviate from the results reported in previous papers.

2) The paper does not compare its method with the state-of-the-art LLM quantization method [1], which enables W4A4 quantization (partially using 8-bit operations) with a PTQ approach.

[1] Zhao, Yilong, et al. "Atom: Low-bit quantization for efficient and accurate llm serving." Proceedings of Machine Learning and Systems 6 (2024): 196-209.

**Questions:**

1) The perplexity and accuracy results of the baseline methods (SQ [1] and OSP [2]) in Table 4 are worse than the figures reported in the original SQ and OSP papers. For instance, the SQ paper reported successful preservation of LLM perplexity after W8A8 quantization (Table 7 of [1]), but Table 4 of this paper shows poor perplexity and accuracy results for SQ. Additionally, the OSP paper claimed successful preservation of LLM perplexity even after INT6 quantization and reported better perplexity results compared to SQ (Table 2 of [2]). Although OSP only evaluated LLaMA-1 and there is no LLaMA-2 data, we can reasonably expect similar trends for LLaMA-2 given that both use GLU-based activation functions. However, Table 4 of this paper shows poor perplexity and accuracy results for OSP, particularly for LLaMA-2-7B. Why are the evaluation results for the previous methods so different?

2) Since the prefix token is retrieved from the calibration set and the threshold alpha is determined from it, will these parameters remain consistent when inferring on a new dataset using the same model?

3) What are the advantages of the proposed method compared to Atom [3] in terms of perplexity/accuracy, latency, or other aspects?

[1] Xiao, Guangxuan, et al. "Smoothquant: Accurate and efficient post-training quantization for large language models." International Conference on Machine Learning. 2023.

[2] Wei, Xiuying, et al. "Outlier suppression+: Accurate quantization of large language models by equivalent and optimal shifting and scaling." arXiv preprint arXiv:2304.09145 (2023).

[3] Zhao, Yilong, et al. "Atom: Low-bit quantization for efficient and accurate llm serving." Proceedings of Machine Learning and Systems 6 (2024): 196-209.

**Limitations:**

The proposed method is limited to GLU-based LLMs.

---

> ### Author Rebuttal · Authors · 2024-08-07
>
> We appreciate the reviewer's insightful comments and the references provided.
>
> ---
>
> **Q1.** The perplexity/accuracy results of the baseline methods deviate from the results reported in previous papers. Why are the evaluation results for the previous methods so different?
>
> **A1.**
> Please refer to Q1 in the general response.
>
> ---
>
> **Q2.** The paper does not compare its method with the state-of-the-art LLM quantization method [1], which enables W4A4 quantization (partially using 8-bit operations) with a PTQ approach. What are the advantages of the proposed method compared to Atom [3] in terms of perplexity/accuracy, latency, or other aspects?
>
> **A2.**
> Please refer to Q3 in the general response.
>
> ---
>
> **Q3.** Since the prefix token is retrieved from the calibration set and the threshold alpha is determined from it, will these parameters remain consistent when inferring on a new dataset using the same model?
>
> **A3.**
> We conducted extra experiments on the LLaMA-2-7B and LLaMA-2-13B models by incorporating various calibration datasets such as WikiText-2, Pile, and PTB to determine the threshold alpha, the target layers of QFeM, and the prefix of QFeP.
> The table below illustrates that for the LLaMA-2-7B model, the alpha ($\alpha$) and the number of excluded layers ($M_{unq}$) remain consistent across datasets, while the prefix tokens are identical in all cases.
> For the LLaMA-2-13B model, the number of excluded layers and most alpha values also exhibit similarity. When evaluating the QFeM and QFeP performance, we found that the results are nearly identical, except the PTB dataset for QFeM.
>
>
> |     Model | **Calibration Dataset** | **$\alpha$** | **$M_{unq}$** | **WikiText-2 (QFeM, ppl$\downarrow$)** |     **Prefix**     | **WikiText-2 (QFeP, ppl$\downarrow$)** |
> |:-----------:|:-------------------:|:--------|:---------:|:----------------------------:|:---------------|:----------------------------:|
> |  LLaMA-2-7B |          C4         |   6.68   |     17    |             5.758            |   [BOS] all .   |             5.758            |
> |  LLaMA-2-7B |      WikiText-2     |   6.68   |     17    |             5.798            |   [BOS] all .   |             5.758            |
> |  LLaMA-2-7B |         Pile        |   6.79   |     15    |             5.768            |   [BOS] all .   |             5.758            |
> |  LLaMA-2-7B |         PTB         |   7.38   |     11    |             5.831            |   [BOS] all .   |             5.758            |
> | LLaMA-2-13B |          C4         |   12.91  |     6     |             5.241            |   [BOS] then ,  |             6.000            |
> |  LLaMA-2-13B  |      WikiText-2     |   37.75  |     4     |             5.291            | [BOS] years the |             6.009            |
> | LLaMA-2-13B |         Pile        |   36.56  |     4     |             5.291            |   [BOS] A the   |             6.000            |
> |  LLaMA-2-13B   |         PTB         |  105.88  |     2     |             5.394            | [BOS] years the |             6.004            |

---

> > ### Comment · Reviewer_BgAS · 2024-08-13
> >
> > Thank you for your response and the additional experiments. You have addressed many of the questions I had, so I have decided to increase my score.

---

> > > ### Author Response · Authors · 2024-08-13
> > >
> > > We greatly appreciate your time and effort in reviewing our rebuttal. Your constructive questions have been instrumental in enhancing the quality of our work.

---

### Author Rebuttal · Authors · 2024-08-07

# Response to all reviewers

We sincerely appreciate all reviewers' thoughtful feedback and constructive suggestions on our paper!

Thanks to valuable comments, our work achieved some breakthroughs and broad contributions:
- Our methods demonstrate effectiveness beyond the W8A8 setting, showing promising results in further low-bit quantization scenarios such as W4A8 and W4A6.
- The high compatibility of our methods enables state-of-the-art (SOTA) LLM quantization methods to acheive performance improvements.

We will gratefully incorporate these improvements into the final version of our paper. Finally, we look forward to further discussion with reviewers!

---

**Q1.** In Table 4, the previous quantization methods (SQ and OSP) show high degradation in evaluation results compared to those reported in their original papers. Why are the evaluation results for the previous methods so different? (Reviewers BgAS, geCr, TX3p)

**A1.** The performance gap is due to the granularity of the activation quantization. Because we identify the activation spikes in token units (Section 3.2), the experiments are mainly on coarse-grained quantization (i.e., **per-tensor quantization**) to examine the impact of activation spikes on a whole tensor (Section 3.3). This may lead to confusing results for the readers. As reviewers commented, baseline methods (SQ and OSP) have reported almost lossless quantization performance for the LLaMA family, as reported in Table 6 of SQ and Table 3 of OSP. As stated in their table captions, they utilze fine-grained **per-token quantization** [1], which we pointed out in line 284 in our paper. To clarify, we expand our Table 4 for both quantization granularities. Please see Table 1 in the attached PDF. The results provide the following insights:
- Although per-token quantization serves nearly lossless quantization for GLU-based LLMs given the W8A8 round-to-nearest (RTN) method, the performance degrades significantly when coarse-grained quantization is applied. Our proposed methods mitigate these quantization errors by addressing the activation spikes, which are responsible for a significant quantization bottleneck.
- With per-token quantization, our methods are still compatible with previous quantization methods (SQ and OSP) and improve quantization performance, especially in the low-bit setting (e.g., W4A4).

---

**Q2.** Extended experimental results regarding low-bit quantization settings (e.g., W4A8, W4A4). (Reviewers TX3p, PpZ5)

**A2.** Thanks for the reviewers' constructive suggestions! We follow the reviewers’ advice and extend our experimental results to include various low-bit quantization scenarios, such as W4A8, W4A6, and W4A4. Please refer to Table 2 in the attached PDF. Our proposed methods consistently improve quantization performance in 4-bit weight quantization, even with 6-bit activation quantization, achieving up to a 29.2% increase in average zero-shot accuracy. However, in W4A4 quantization, the limited bitwidth for activations ruins the functionality of LLM with coarse-grained quantization. From this observation, we conclude that fine-grained activation quantization is necessary for extremely low-bit cases, such as 4-bit.

---

**Q3.** Additional comparisons with state-of-the-art (SOTA) LLM quantization methods. (Reviewers BgAS, PpZ5)

**A3.** We truly appreciate the references that reviewers recommended for the SOTA LLM quantization methods. Because our methods are simple to integrate and orthogonal to LLM quantization techniques (e.g., custom matrix multiplication), one can directly plug in our QFeM or QFeP, or even both. Among the recommended SOTA methods, we tested Atom [2] and OmniQuant [3] with our methods by evaluating the perplexity of WikiText-2 and C4 datasets.

| Model       | #Bits | Method        | WikiText-2 |   C4   |
|-------------|-------|---------------|:----------:|:------:|
| LLaMA-2-7B  | FP16  | -             |    5.268   |  7.013 |
| LLaMA-2-7B  | W4A4  | Atom          |    5.710   |  7.601 |
| LLaMA-2-7B  | W4A4  | $\quad$+QFeM  |    5.634   |  7.538 |
| LLaMA-2-7B  | W4A4  | $\quad$+QFeP         |    5.685   |  7.493 |
| LLaMA-2-7B  | W4A4  | $\quad$+QFeM+QFeP    |    **5.607**   |  **7.442** |
| LLaMA-2-7B  | W4A4  | OmniQuant    |   14.208   | 19.005 |
| LLaMA-2-7B  | W4A4  | $\quad$+QFeM         |    9.483   | 13.569 |
| LLaMA-2-7B  | W4A4  | $\quad$+QFeP         |   11.926   | 15.796 |
| LLaMA-2-7B  | W4A4  | $\quad$+QFeM+QFeP    |    **8.818**   | **12.244** |
| LLaMA-2-13B | FP16  | -             |    4.789   |  6.518 |
| LLaMA-2-13B | W4A4  | Atom          |    5.081   |  6.878 |
| LLaMA-2-13B | W4A4  | $\quad$+QFeM         |    **5.071**   |  **6.854** |
| LLaMA-2-13B | W4A4  | $\quad$+QFeP         |    5.089   |  6.872 |
| LLaMA-2-13B | W4A4  | $\quad$+QFeM+QFeP    |    5.073   |  6.855 |
| LLaMA-2-13B | W4A4  | OmniQuant     |   10.416   | 14.103 |
| LLaMA-2-13B | W4A4  | $\quad$+QFeM         |    **9.499**   | 12.348 |
| LLaMA-2-13B | W4A4  | $\quad$+QFeP         |   10.808   | 13.584 |
| LLaMA-2-13B | W4A4  | $\quad$+QFeM+QFeP    |    9.901   | **12.294** |

Note that we follow their original implementation and parameter settings (e.g., fine-grained group quantization in Atom) during evaluation. As highlightened in bold in above table, our methods are compatible with SOTA quantization methods (Atom and OmniQuant) and enhance their quantization performance, especially for OmniQuant (and W4A4 OSP in Q1).

---

**references:**

[1] Yao, Zhewei, et al. "Zeroquant: Efficient and affordable post-training quantization for large-scale transformers." Advances in Neural Information Processing Systems 35 (2022): 27168-27183.

[2] Zhao, Yilong, et al. "Atom: Low-bit quantization for efficient and accurate llm serving." Proceedings of Machine Learning and Systems 6 (2024): 196-209.

[3] Shao, Wenqi, et al. "Omniquant: Omnidirectionally calibrated quantization for large language models." arXiv preprint arXiv:2308.13137 (2023).

---

### Decision · Program_Chairs · 2024-09-25

**Decision:**

Reject

**Comment:**

The reviewers' criticism and recommendations are very consistent. One major issue was the lack of comparison to state-of-the-art LLM and the strong resemblance to a previously researched methods. Apparently, the authors were able to address these concerns in the rebuttal. Finally, all reviewers recommend borderline acceptance.
After reading the paper I also think that -although the technical innovation of the paper is moderate- it is a good paper or sufficient interest to the NeurIPS community. However, after a discussion with the SAC, I change my recommendation to reject.